# Piecewise-Velocity Model for Learning Continuous-time Dynamic Node Representations

**Abdulkadir Çelikkanat, Nikolaos Nakis, Morten Mørup**
Technical University of Denmark
Kongens Lyngby 2800, Denmark
`abce@dtu.dk nnak@dtu.dk mmor@dtu.dk`

## Abstract

Networks have become indispensable and ubiquitous structures in many fields to model the interactions among different entities, such as friendship in social networks or protein interactions in biological graphs. A major challenge is to understand the structure and dynamics of these systems. Although networks evolve through time, most existing graph representation learning methods target only static networks. Whereas approaches have been developed for the modeling of dynamic networks, there is a lack of efficient continuous time dynamic graph representation learning methods that can provide accurate network characterization and visualization in low dimensions while explicitly accounting for prominent network characteristics such as homophily and transitivity. In this paper, we propose the Piecewise-VElocity Model (PiVEM) for the representation of continuous-time dynamic networks. It learns dynamic embeddings in which the temporal evolution of nodes is approximated by piecewise linear interpolations based on a latent distance model with piecewise constant node-specific velocities. The model allows for analytically tractable expressions of the associated Poisson process likelihood with scalable inference invariant to the number of events. We further impose a scalable Kronecker structured Gaussian Process prior to the dynamics accounting for community structure, temporal smoothness, and disentangled (uncorrelated) latent embedding dimensions optimally learned to characterize the network dynamics. We show that PiVEM can successfully represent network structure and dynamics in ultra-low two-dimensional embedding spaces. We further extensively evaluate the performance of the approach on various networks of different types and sizes and find that it outperforms existing relevant state-of-art methods in downstream tasks such as link prediction. In summary, PiVEM enables easily interpretable dynamic network visualizations and characterizations that can further improve our understanding of the intrinsic dynamics of time-evolving networks.

## 1 Introduction

With technological advancements in data storage and production systems, we have witnessed the massive growth of graph (or network) data in recent years, with many prominent examples, including social, technological, and biological networks from diverse disciplines [1]. They propose an exquisite way to store and represent the interactions among data points and machine learning techniques on graphs have thus gained considerable attention to extract meaningful information from these complex systems and perform various predictive tasks. In this regard, *Graph Representation Learning (GRL)* techniques have become a cornerstone in the field through their exceptional performance in many downstream tasks such as node classification and edge prediction. Unlike the classical techniques relying on the extraction and design of handcrafted feature vectors peculiar to given networks, GRL approaches aim to design algorithms that can automatically learn features optimally preserving various characteristics of networks in their induced latent space.

A. Celikkanat et al., Piecewise-Velocity Model for Learning Continuous-time Dynamic Node Representations.
*Proceedings of the First Learning on Graphs Conference (LoG 2022)*, PMLR 198, Virtual Event, December 9–12, 2022.

Many networks evolve through time and are liable to modifications in structure with newly arriving nodes or emerging connections, the GRL methods have primarily addressed static networks, in other words, a snapshot of the networks at a specific time. However, recent years have seen increasing efforts toward modeling dynamic complex networks, see also [2] for a review. Whereas most approaches have concentrated their attention on discrete-time temporal networks, which have built upon a collection of time-stamped networks (c.f. [2–10]) modeling of networks in continuous time has also been studied (c.f. [11–14]). These approaches have been based on latent class [3, 4, 11–13] and latent feature modeling approaches [2, 5–10, 14], including advanced dynamic graph neural network representations [15, 16].

Although these procedures have enabled the characterization of evolving networks for downstream tasks such as link prediction and node classification, existing dynamic latent feature models are either in discrete time or do not explicitly account for network homophily and transitivity in terms of their latent representations. Whereas latent class models typically provide interpretable representations at the level of groups, latent feature models in general rely on high-dimensional latent representations that are not easily amenable to visualization and interpretation. A further complication of most existing dynamic modeling approaches is their scaling typically growing in complexity by the number of observed events and number of network dyads.

This work addresses the embedding problem of nodes in a continuous-time latent space and seeks to model network interaction patterns using low-dimensional representations accurately. We model the node interactions with Nonhomogeneous Poisson Point Processes whose densities are defined based on the relative distances among the node trajectories in the latent space. The node movements are characterized by node-specific piecewise velocity vectors, such that each node acquires a dynamic representation pursuing a continuous path in the latent space throughout the timeline. The main contributions of the paper can be summarized as follows:

- We propose a novel scalable GRL method, the Piecewise-VElocity Model (PiVeM), to flexibly learn continuous-time dynamic node representations. The temporal evolutions of networks are represented by piecewise linear motions of the nodes' embeddings in the latent space.

- We present a framework balancing the trade-off between the smoothness of node trajectories in the latent space and model capacity accounting for the temporal evolution.

- We show that the PiVeM can embed nodes accurately in very low dimensional spaces, i.e., $D = 2$, such that it serves as a dynamic network visualization tool facilitating human insights into networks' complex, evolving structures.

- The performance of the introduced approach is extensively evaluated in various downstream tasks, such as network reconstruction and link prediction. We show that it outperforms well-known baseline methods on a wide range of datasets. Besides, we propose an efficient model optimization strategy enabling the PiVeM to scale to large networks.

**Source code and other materials.** The datasets, the implementation, and all the generated animations can be found at the address: `https://abdcelikkanat.github.io/projects/pivem/`.

## 2 Related Work

The work on dynamic modeling of complex networks has spurred substantial attention in recent years and covers approaches for the modeling of dynamic structures at the level of groups (i.e., latent class models) and dynamic representation learning approaches based on latent feature models, including graph neural networks (GNNs). Whereas most attention has been given to discrete-time dynamic networks, a substantial body of work has also covered continuous-time modeling, as outlined below.

**Dynamic Latent Class Models.** Initial efforts for modeling continuously evolving networks has combined latent class models defined by the stochastic block models [17, 18] with Hawkes processes [19, 20]. In the work of [11], co-dependent (through time) Hawkes processes were combined with the Infinite Relational Model [21] (Hawkes IRM), yielding a non-parametric Bayesian approach capable of expressing reciprocity between inferred groups of actors. A drawback of such a model is the computational cost of the imposed Markov-chain Monte-Carlo optimization, as well as, its limitation on modeling only reciprocation effects. Scalability issues were addressed in [12] via the Block Hawkes Model (BHM), which utilizes variational inference and simplifies the Hawkes IRM model by associating only the inferred block structure pairs with a univariate point process. Recently,

the BHM model was extended to decoupling interactions between different pairs of nodes belonging to the same block pair, through the use of independent univariate Hawkes processes, defining the Community Hawkes Independent Pairs model [13]. Whereas the above works have been based on continuous time modeling of dynamic networks, the dynamic-IRM (dIRM) of [3] focused on the modeling of discrete-time networks by inducing an infinite Hidden Markov Model (IHMM) to account for transitions over time of nodes between communities. In [4], a dynamic hierarchical block model was proposed based on the modeling of change points admitting dynamic node relocation within a Gibbs fragmentation tree. Despite the various advantages of such models, networks are constrained to be regarded and analyzed at a block level which in many cases is restrictive.

**Dynamic Latent Feature Models.** Prominent works around node-level representations of continuous-time networks [22, 23] have originally considered feature propagation within the discrete time network topology [5] or extended the random-walk frameworks [6, 7] to the temporal case yielding the CTDNE [24] model. CTDNE provides a single temporal-aware node embedding, meaning that network and node evolution are unable to be visualized and explored. A more flexible approach was designed in [15] (DYREP), where temporal node embeddings are learned under a so-called latent mediation process, combining an association process describing the dynamics of the network with a communication process describing the dynamics on the network. It uses deep recurrent architectures to parameterize the intensity function of the point process, and thus the embedding space suffers from a lack of explainability. HTNE [25] introduces a model utilizing a Hawkes process relying on node embeddings. Unlike many approaches concentrating only on the structural modifications occurring between nodes, MMDNE [26] explicitly considers such pairwise micro, and network scale macro dynamics and uses a temporal node representation learning algorithm relying on a temporal attention point process. Graph neural networks (GNNs) can be extended to the analysis of continuous networks via the Temporal Graph Network (TGN) [16] where the classical encoder-decoder architecture is coupled with a memory cell.

In the context of latent feature dynamic network models, Gaussian Processes (GP) have been used to characterize the smoothness of the temporal dynamics. This includes the discrete-time dynamic models considered in [8] in which latent factors were endowed a GP prior based on radial basis kernels imposing temporal smoothness within the latent representation. The approach was extended in [9] to impose stochastic differential equations for the evolution of latent factors. In [14], GPs were used for the modeling of continuous-time dynamic networks based on Poisson and Hawkes processes, including exogenous as well as endogenous features specified by a radial basis function prior.

Latent Distance Models (LDM) [27] have recently been shown to outperform prominent GRL methods utilizing very-low dimensions in the static case [28, 29]. LDMs for temporal networks have been mostly studied in the discrete case [2], considering mainly diffusion dynamics to make predictions, as firstly studied in [30] and extended with popularity and activity effects [10]. While all these models express homophily and transitivity in the dynamic case, they fail to account for continuous dynamics.

Our work is inspired by these previous approaches for the modeling of dynamic complex networks. Specifically, we make use of the latent distance model formulation to account for homophily and transitivity, the Poisson Process for the characterization of continuous-time dynamics, and a Gaussian Process prior based on the radial-basis-function kernel to account for temporal smoothness within the latent representation. Inspired by latent class models, we further impose a structured low-rank representation of nodes based on soft-assigning nodes to communities exhibiting similar temporal dynamics. Notably, we exploit how LDMs as opposed to GNN approaches in general, can provide easily interpretable yet accurate network representations in ultra-low dimensional spaces ($D = 2$), facilitating accurate dynamic network visualization and interpretation.

## 3 Proposed Approach

Our main objective is to represent every node of a given network, $\mathcal{G} = (\mathcal{V}, \mathcal{E})$, into a low-dimensional metric space, $(\mathsf{X}, d_\mathsf{X})$, in which the pairwise node proximities will be characterized by their distances in a continuous-time latent space (Objective 3.1). Since we address the continuous-time dynamic networks, the interactions among nodes through time can vary, with new links appearing or disappearing at any time. More precisely, we will presently consider undirected continuous-time networks:

**Definition 3.1.** A *continuous-time dynamic undirected graph* on a time interval $\mathcal{I}_T := [0, T]$ is an ordered pair $\mathcal{G} = (\mathcal{V}, \mathcal{E})$ where $\mathcal{V} = \{1, \ldots, N\}$ is a set of nodes and $\mathcal{E} \subseteq \{\{i, j, t\} \in \mathcal{V}^2 \times \mathcal{I}_T | 1 \le i < j \le N\}$ is a set of *events* or *edges*.

We will use the symbol, $N$, to denote the number of nodes in the vertex set and $\mathcal{E}_{ij}[t_l, t_u] \subseteq \mathcal{E}$ to indicate the set of edges between nodes $i$ and $j$ occurring on the interval $[t_l, t_u] \subseteq \mathcal{I}_T$.

### 3.1 Nonhomogeneous Poisson Point Processes

The *Poisson Point Processes (PPP)s* are one of the natural choices widely used to model the number of random events occurring in time or the locations in a spatial space. PPPs are parameterized by a quantity known as the rate or the intensity indicating the average density of the points in the underlying space of the Poisson process. If the intensity depends on the time or location, the point process is called *Nonhomogeneous PPP* (Defn. 3.2), and it is typically adapted for applications in which the event points are not uniformly distributed [31].

**Definition 3.2.** [Nonhomogeneous PPP] A counting process $\{M(t), t \geq 0\}$ is called a *nonhomogeneous Poisson process* with *intensity function* $\lambda(t)$, $t \geq 0$ if **(i)** $M(0) = 0$, **(ii)** $M(t)$ has independent increments: i.e., $\big(M(t_1) - M(t_0)\big), \ldots, \big(M(t_B) - M(t_{B-1})\big)$ are independent random variables for each $0 \leq t_0 < \cdots < t_B$, and **(iii)** $M(t_u) - M(t_l)$ is Poisson distributed with mean $\int_{t_l}^{t_u} \lambda(t)dt$.

In this paper, we consider continuous-time dynamic undirected networks such that the events (or links/edges) among nodes can occur at any point in time. As we will examine in the following sections, these interactions do not necessarily exhibit any recurring characteristics; instead, they vary over time in many real networks. In this regard, we assume that the number of links, $M[t_l, t_u]$, between a pair of nodes $(i, j) \in \mathcal{V}^2$ ($i < j$ since the graph is undirected) follows a nonhomogeneous Poisson point process (NHPP) with intensity function $\lambda_{ij}(t)$ on the time interval $[t_l, t_u)$, and the log-likelihood function can be written by

$$\mathcal{L}(\Omega) := \log p(\mathcal{G}|\Omega) = \sum_{\substack{i<j \\ i,j\in\mathcal{V}}} \left( \sum_{e_{ij}\in\mathcal{E}_{ij}} \log \lambda_{ij}(e_{ij}) - \int_0^T \lambda_{ij}(t)dt \right) \tag{1}$$

where $\mathcal{E}_{i,j} \subseteq \mathcal{E}[0, T]$ is the set of links of node pair $(i, j) \in \mathcal{V}^2$ on the timeline $\mathcal{I}_T := [0, T]$ for a network $\mathcal{G} = (\mathcal{V}, \mathcal{E})$, and $\Omega = \{\lambda_{ij}\}_{1 \leq i < j \leq N}$ indicates the set of intensity functions.

### 3.2 Problem Formulation

Without loss of generality, it can be assumed that the timeline starts from $0$ and is bounded by $T \in \mathbb{R}^+$. Since the interactions among nodes can occur at any time point on $\mathcal{I}_T = [0, T]$, we would like to identify an accurate continuous-time node representation $\{r(i, t)\}_{(i,t)\in\mathcal{V}\times\mathcal{I}_T}$ defined using a low-dimensional latent space $\mathbb{R}^D$ ($D \ll N$) where $\mathbf{r} : \mathcal{V} \times \mathcal{I}_T \to \mathbb{R}^D$ is a map indicating the embedding or representation of node $i \in \mathcal{V}$ at time point $t \in \mathcal{I}_T$. We define our objective more formally as follows:

**Objective 3.1.** *Let $\mathcal{G} = (\mathcal{V}, \mathcal{E})$ be a continuous-time dynamic network and $\boldsymbol{\lambda}^* : \mathcal{V}^2 \times \mathcal{I}_T \longrightarrow \mathbb{R}$ be an unknown intensity function of a nonhomogeneous Poisson point process. For a given metric space $(\mathsf{X}, d_\mathsf{X})$, our purpose is to learn a function or representation $\mathbf{r} : \mathcal{V} \times \mathcal{I}_T \to \mathsf{X}$ satisfying*

$$\frac{1}{(t_u - t_l)} \int_{t_l}^{t_u} \psi^+\Big(d_\mathsf{X}\big(\mathbf{r}(i, t), \mathbf{r}(j, t)\big)\Big) dt \approx \frac{1}{(t_u - t_l)} \int_{t_l}^{t_u} \boldsymbol{\lambda}^*(i, j, t)dt \tag{2}$$

*for a continuous function $\psi^+ : \mathbb{R} \to \mathbb{R}^+$ for all $(i, j) \in \mathcal{V}^2$ pairs, and for every interval $[t_l, t_u] \subseteq \mathcal{I}_T$.*

In this work, we consider the Euclidean metric on a $D$-dimensional real vector space, $\mathsf{X} := \mathbb{R}^D$ and the embedding of node $i \in \mathcal{V}$ at time $t \in \mathcal{I}_T$ will be denoted by $\mathbf{r}_i(t) \in \mathbb{R}^D$.

### 3.3 PIVEM: Piecewise-Velocity Model For Learning Continuous-time Embeddings

We learn continuous-time node representations by employing the canonical exponential link-function defining the intensity function as

$$\lambda_{ij}(t) := \exp\Big(\beta_i + \beta_j - ||\mathbf{r}_i(t) - \mathbf{r}_j(t)||^2\Big) \tag{3}$$

where $\mathbf{r}_i(t) \in \mathbb{R}^D$ and $\beta_i \in \mathbb{R}$ denote the embedding vector at time $t$ and the bias term of node $i \in \mathcal{V}$, respectively. Importantly, for a pair of nodes, we would like to have embeddings close enough to

each other when they have high interactions during a particular time interval and far away from each other if they have less or no links. For given bias terms, it can be seen by the following Lemma 3.1, that the definition of the intensity function provides a guarantee for our goal given in Equation (2).

**Lemma 3.1.** *For given fixed bias terms* $\{\beta_i\}_{i \in \mathcal{V}}$, *the node embeddings,* $\{\mathbf{r}_i(t)\}_{i \in \mathcal{V}}$, *learned by the objective given in Equation 1 within a bounded set of radius* $R_t$ *during a time interval* $[t_l, t_u]$ *satisfy*

$$\log\left(\frac{(t_u - t_l)}{-\log p_{ij}^0}\right) + (\beta_i + \beta_j) \le \frac{1}{(t_u - t_l)}\int_{t_l}^{t_u} ||\mathbf{r}_i(t) - \mathbf{r}_j(t)||^2 dt \le \log\left(\frac{(t_u - t_l)}{-\log(1 - p_{ij}^>)}\right) + (\beta_i + \beta_j) + R_t$$

*where* $p_{ij}^0$ *and* $p_{ij}^>$ *are the probabilities of having zero and more than zero links for the nodes* $i$ *and* $j$.

*Proof.* Please see the appendix for the proof. □

When we take a look at the lower bound provided in Lemma 3.1 for a pair of nodes having no links $(p_{ij}^0 \to 1)$ within a particular time interval $[t_l, t_u]$, the lower bound converges to infinity, so nodes are positioned in distant locations. Similarly, the upper bound provides us an intuition about the position of the nodes for the case in which they exhibit a high number of interactions ($p_{ij}^> \to 1$). The log term squeezes the distance, and the nodes are forced to be positioned close to each other. As we will see in the following parts, we restrict the node representations within a bounded region of radius $R_t$ by imposing a prior function in order to restrain the movements in the latent space. We utilize the squared Euclidean distance in Equation (3), which is not a metric, but we presently impose it as a distance [29, 32] for computational convenience, see Lemma A.4 in Appendix.

Notably, constraining the approximation of the unknown intensity function by a metric space imposes the homophily property (i.e., similar nodes in the graph are placed close to each other in embedding space). It can also be seen that the transitivity property holds up to some extent (i.e., if node $i$ is similar to $j$ and $j$ similar to $k$, then $i$ should also be similar to $k$) since we can bound the squared Euclidean distance [29, 33]. Note that the bias terms $\{\beta_i\}_{i \in \mathcal{V}}$ are responsible for the node-specific effects such as degree heterogeneity [28, 33], and they provide additional flexibility to the model by acting as scaling factor for the corresponding nodes so that, for instance, a hub node might have a high number of interactions simultaneously without getting close to the others in the latent space.

Since our primary purpose is to learn continuous-time node representations in a latent space, we define the representation of node $i \in \mathcal{V}$ at time $t \in \mathcal{I}_T$ based on a linear model by $\mathbf{r}_i(t) := \mathbf{x}_i^{(0)} + \mathbf{v}_i t$. Here, $\mathbf{x}_i^{(0)}$ can be considered as the initial position and $\mathbf{v}_i$ the velocity of the corresponding node. However, the linear model provides a minimal capacity for tracking the nodes and modeling their representations. Therefore, we reinterpret the given timeline $\mathcal{I}_T := [0, T]$ by dividing it into $B$ equally-sized bins, $[t_{b-1}, t_b)$, $(1 \le b \le B)$ such that $[0, T] = [0, t_1) \cup \cdots \cup [t_{B-1}, t_B]$ where $t_0 := 0$ and $t_B := T$. By applying the linear model for each subinterval, we obtain a piecewise linear approximation of general intensity functions strengthening the models' capacity. As a result, we can write the position of node $i$ at time $t \in \mathcal{I}_T$ as follows:

$$\mathbf{r}_i(t) := \mathbf{x}_i^{(0)} + \Delta_B \mathbf{v}_i^{(1)} + \Delta_B \mathbf{v}_i^{(2)} + \cdots + (t \bmod(\Delta_B))\mathbf{v}_i^{\left(\lfloor t/\Delta_B \rfloor + 1\right)} \tag{4}$$

where $\Delta_B$ indicates the bin widths, $T/B$, and $\bmod(\cdot)$ is the modulo operation used to compute the remaining time. Note that the piece-wise interpretation of the timeline allows us to track better the path of the nodes in the embedding space, and it can be seen by Theorem 3.2 that we can obtain more accurate trails by augmenting the number of bins.

**Theorem 3.2.** *Let* $\mathbf{f}(t) : [0, T] \to \mathbb{R}^D$ *be a continuous embedding of a node. For any given* $\epsilon > 0$, *there exists a continuous, piecewise-linear node embedding,* $\mathbf{r}(t)$, *satisfying* $||\mathbf{f}(t) - \mathbf{r}(t)||_2 < \epsilon$ *for all* $t \in [0, T]$ *where* $\mathbf{r}(t) := \mathbf{r}^{(b)}(t)$ *for all* $(b-1)\Delta_B \le t < b\Delta_B$, $\mathbf{r}(t) := \mathbf{r}^{(B)}(t)$ *for* $t = T$ *and* $\Delta_B = T/B$ *for some* $B \in \mathbb{N}^+$.

*Proof.* Please see the appendix for the proof. □

**Prior probability.** In order to control the smoothness of the motion in the latent space, we employ a Gaussian Process (GP) [34] prior over the initial position $\mathbf{x}^{(0)} \in \mathbb{R}^{N \times D}$ and velocity vectors $\mathbf{v} \in \mathbb{R}^{B \times N \times D}$. Hence, we suppose that $\text{vect}(\mathbf{x}^{(0)}) \oplus \text{vect}(\mathbf{v}) \sim \mathcal{N}(\mathbf{0}, \mathbf{\Sigma})$ where $\mathbf{\Sigma} := \lambda^2(\sigma_\Sigma^2 \mathbf{I} + \mathbf{K})$ is the covariance matrix with a scaling factor $\lambda \in \mathbb{R}$. We utilize, $\sigma_{\Sigma \in \mathbb{R}}$, to denote the noise

of the covariance, and $\text{vect}(\mathbf{z})$ is the vectorization operator stacking the columns to form a single vector. To reduce the number of parameters of the prior and enable scalable inference, we define $\mathbf{K}$ as a Kronecker product of three matrices $\mathbf{K} := \mathbf{B} \otimes \mathbf{C} \otimes \mathbf{D}$ respectively accounting for temporal-, node-, and dimension specific covariance structures. Specifically, we define $\mathbf{B} := \big[ c_{\mathbf{x}^0} \big] \oplus \big[ \exp(-(c_b - \tilde{c}_{\tilde{b}})^2 / 2\sigma_{\mathbf{B}}^2) \big]_{1 \le b, \tilde{b} \le B}$ is a $(B+1) \times (B+1)$ matrix intending to capture the smoothness of velocities across time-bins where $c_b = \frac{t_{b-1} + t_b}{2}$ is the center of the corresponding bin, and the matrix is constructed by combining the radial basis function kernel (RBF) with a scalar term $c_{\mathbf{x}^0}$ corresponding to the initial position being decoupled from the structure of the velocities. The node specific matrix, $\mathbf{C} \in \mathbb{R}^{N \times N}$, is constructed as a product of a low-rank matrix $\mathbf{C} := \mathbf{Q}\mathbf{Q}^\top$ where the row sums of $\mathbf{Q} \in \mathbb{R}^{N \times k}$ equals to $1$ $(k \ll N)$, and it aims to extract covariation patterns of the motion of the nodes. Finally, we simply set the dimensionality matrix to the identity: $\mathbf{D} := \mathbf{I} \in \mathbb{R}^{D \times D}$ in order to have uncorrelated dimensions.

To sum up, we can express our objective relying on the piecewise velocities with the prior as follows:

$$\hat{\Omega} = \arg\max_{\Omega} \sum_{\substack{i<j \\ i,j \in \mathcal{V}}} \left( \sum_{e_{ij} \in \mathcal{E}_{ij}} \log \lambda_{ij}(e_{ij}) - \int_0^T \lambda_{ij}(t)dt \right) + \log \mathcal{N} \left( \left[ \begin{array}{c} \mathbf{x}^{(0)} \\ \mathbf{v} \end{array} \right]; \mathbf{0}, \mathbf{\Sigma} \right) \quad (5)$$

where $\Omega = \{\boldsymbol{\beta}, \mathbf{x}^{(0)}, \mathbf{v}, \sigma_\Sigma, \sigma_\mathbf{B}, c_{\mathbf{x}^0}, \mathbf{Q}\}$ is the set of hyper-parameters, and $\lambda_{ij}(t)$ is the intensity function as defined in Equation (3) based on the node embeddings, $\mathbf{r}_i(t) \in \mathbb{R}^D$.

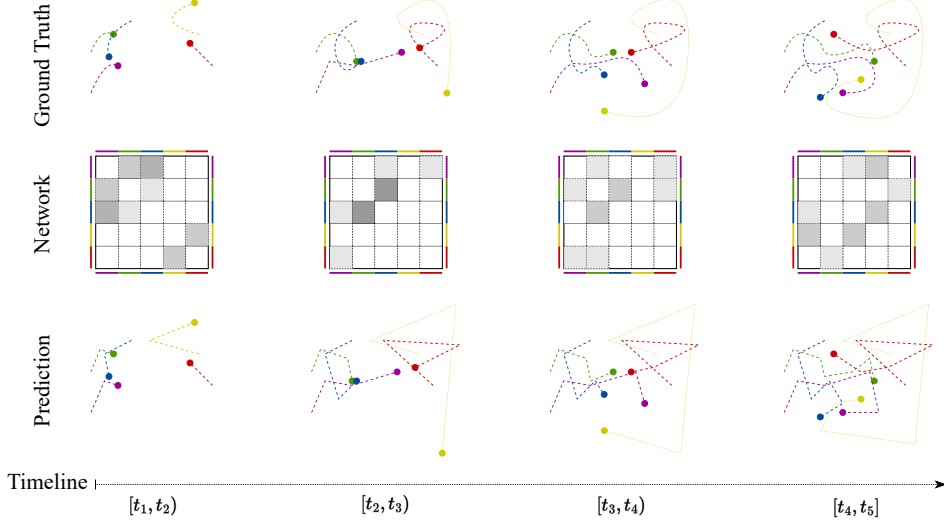

**Figure 1:** Illustrative comparison of the ground-truth embeddings, the adjacency matrices here for illustrative purposes constructed based on aggregating the links appearing within the corresponding time intervals and learned node representations.

We provide the general overview of the PIVEM method in Figure 1. The first row shows how the ground truth node embeddings evolve through time, and the dashed curves in the latent space show the paths they have followed. The middle row represents the adjacency matrices of the network constructed by aggregating the links occurring within the corresponding time intervals $[t_{init}, t_{last}]$ for illustrative purposes (notably, the model operates in continuous time and accounts for the temporal position of each edge). Each entry of the adjacency matrices is shaded with respect to the number of links in the intervals, so darker regions represent a higher number of links. Finally, the last row illustrates the learned representations and their motion histories in the latent space.

### 3.4 Optimization

Our objective given in Equation (5) is not a convex function, so the learning strategy that we follow is of great significance in order to escape from the local minima and for the quality of the representations. We start by randomly initializing the model's hyper-parameters from $[-1, 1]$ except for the velocity

tensor, which is set to $0$ at the beginning. We adapt the sequential learning strategy in learning these parameters. In other words, we first optimize the initial position and bias terms together, $\{\mathbf{x}^{(0)}, \boldsymbol{\beta}\}$, for a given number of epochs; then, we include the velocity tensor, $\{\mathbf{v}\}$, in the optimization process and repeat the training for the same number of epochs. Finally, we add the prior parameters and learn all model hyper-parameters together. We have employed *Adam optimizer* [35] with learning rate $0.1$.

**Computational issues and complexity.** Note that we need to evaluate the log-intensity term in Equation (5) for each $(i, j) \in \mathcal{V}^2$ and event time $e_{ij} \in \mathcal{E}_{ij}$. Therefore, the computational cost required for the whole network is bounded by $\mathcal{O}\left(|\mathcal{V}|^2|\mathcal{E}|\right)$. However, we can alleviate the computational cost by pre-computing certain coefficients at the beginning of the optimization process so that the complexity can be reduced to $\mathcal{O}\left(|\mathcal{V}|^2 B\right)$. We also have an explicit formula for the computation of the integral term since we utilize the squared Euclidean distance so that it can be computed in at most $\mathcal{O}(|\mathcal{V}|^2)$ operations. Instead of optimizing the whole network at once, we apply a batching strategy over the set of nodes in order to reduce the memory requirements. As a result, we sample $\mathcal{S}$ nodes for each epoch. Hence, the overall complexity for the log-likelihood function is $\mathcal{O}\left(\mathcal{S}^2 B \mathcal{I}\right)$ where $\mathcal{I}$ is the number of epochs and $\mathcal{S} \ll |\mathcal{V}|$. Similarly, the prior can be computed in at most $\mathcal{O}(B^3 D^3 K^2 \mathcal{S})$ operations by using various algebraic properties such as *Woodbury matrix identity* and *Matrix Determinant lemma* [36]. To sum up, the complexity of the proposed approach is $\mathcal{O}(B\mathcal{S}^2\mathcal{I} + B^3 D^3 K^2 \mathcal{S}\mathcal{I})$ (Please see the appendix for the derivations and other details).

## 4  Experiments

In this section, we extensively evaluate the performance of the proposed PIecewise-VElocity Model with respect to the well-known baselines in challenging tasks over various datasets of sizes and types. We consider all networks as undirected, and the event times of links are scaled to the interval $[0, 1]$ for the consistency of experiments. We use the finest granularity level of the given input timestamps, such as seconds and milliseconds. We provide a brief summary of the networks below, but more details and various statistics are reported in Table 4 in the appendix. For all the methods, we learn node embeddings in two-dimensional space ($D = 2$) since one of the objectives of this work is to produce dynamic node embeddings facilitating human insights into a complex network.

**Experimental Setup.** We first split the networks into two sets, such that the events occurring in the last $10\%$ of the timeline are taken out for the prediction. Then, we randomly choose $10\%$ of the node pairs among all possible dyads in the network for the graph completion task, and we ensure that each node in the residual network contains at least one event keeping the number of nodes fixed. If a pair of nodes only contains events in the prediction set and if these nodes do not have any other links during the training time, they are removed from the networks.

For conducting the experiments, we generate the labeled dataset of links as follows: For the positive samples, we construct small intervals of length $2 \times 10^{-3}$ for each event time (i.e., $[e - 10^{-3}, e + 10^3]$ where $e$ is an event time). We randomly sample an equal number of time points and corresponding node pairs to form negative instances. If a sampled event time is not located inside the interval of a positive sample, we follow the same strategy to build an interval for it, and it is considered a negative instance. Otherwise, we sample another time point and a dyad. Note that some networks might contain a very high number of links, which leads to computational problems for these networks. Therefore, we subsample $10^4$ positive and negative instances if they contain more than this.

**Synthetic networks.** We generate two artificial networks in order to evaluate the behavior of the models in controlled experimental settings. **(i)** *Synthetic($\pi$)* is sampled from the prior distribution stated in Subsection 3.2. The hyper-parameters, $\boldsymbol{\beta}$, $K$ and $B$ are set to $\mathbf{0}$, $20$ and $100$, respectively. **(ii)** *Synthetic($\mu$)* is constructed based on the temporal block structures. The timeline is divided into $10$ sub-intervals, and the nodes are randomly split into $20$ groups for each interval. The links within each group are sampled from the Poisson distribution with the constant intensity of $5$.

**Real networks.** The **(iii)** *Hypertext* network [37] was built on the radio badge records showing the interactions of the conference attendees for 2.5 days, and each event time indicates 20 seconds of active contact. Similarly, **(iv)** the *Contacts* network [38] was generated concerning the interactions of the individuals in an office environment. **(v)** *Forum* [39] is comprised of the activity data of university students on an online social forum system. **(vi)** *College* [40] indicates the private messages among the students on an online social platform. Finally, **(vii)** *Email* [41] was constructed based on the exchanged e-mail information among the members of European research institutions.

**Table 1:** The performance evaluation for the network reconstruction experiment over various datasets.

| | Synthetic($\pi$) | | Synthetic($\mu$) | | College | | Contacts | | Email | | Forum | | Hypertext | |
|---|---|---|---|---|---|---|---|---|---|---|---|---|---|---|
| | ROC | PR | ROC | PR | ROC | PR | ROC | PR | ROC | PR | ROC | PR | ROC | PR |
| LDM | .563 | .539 | .669 | .642 | **.951** | .944 | .860 | .835 | .954 | .948 | **.909** | .897 | .818 | .797 |
| NODE2VEC | .519 | .507 | .503 | .509 | .711 | .655 | .812 | .756 | .853 | .828 | .677 | .619 | .696 | .648 |
| CTDNE | .613 | .580 | .539 | .544 | .661 | .622 | .787 | .760 | .854 | .840 | .657 | .622 | .725 | .725 |
| HTNE | .614 | .591 | .599 | .571 | .721 | .683 | .846 | .823 | .871 | .867 | .723 | .691 | .775 | .787 |
| MMDNE | .582 | .565 | .600 | .576 | .725 | .692 | .844 | .825 | .867 | .863 | .737 | .712 | .778 | .787 |
| PIVEM | **.762** | **.713** | **.905** | **.869** | .948 | **.948** | **.938** | **.938** | **.978** | **.977** | .907 | **.902** | **.830** | **.823** |

**Table 2:** The performance evaluation for the network completion experiment over various datasets.

| | Synthetic($\pi$) | | Synthetic($\mu$) | | College | | Contacts | | Email | | Forum | | Hypertext | |
|---|---|---|---|---|---|---|---|---|---|---|---|---|---|---|
| | ROC | PR | ROC | PR | ROC | PR | ROC | PR | ROC | PR | ROC | PR | ROC | PR |
| LDM | .535 | .529 | .646 | .631 | .931 | .926 | .836 | .799 | .948 | .942 | .863 | .858 | .761 | **.738** |
| NODE2VEC | .519 | .511 | .747 | .677 | .685 | .637 | .787 | .744 | .818 | .777 | .635 | .592 | .596 | .588 |
| CTDNE | .608 | .573 | .531 | .539 | .601 | .556 | .752 | .703 | .831 | .812 | .568 | .539 | .554 | .537 |
| HTNE | .605 | .583 | .573 | .557 | .673 | .651 | .792 | .759 | .853 | .834 | .596 | .581 | .602 | .633 |
| MMDNE | .587 | .570 | .592 | .571 | .677 | .662 | .819 | .811 | .844 | .829 | .596 | .570 | .587 | .614 |
| PIVEM | **.750** | **.696** | **.874** | **.851** | **.935** | **.934** | **.873** | **.864** | **.951** | **.953** | **.879** | **.875** | **.770** | .712 |

**Baselines.** We compare the performance of our method with five baselines. We include (i) LDM with Poisson rate with node-specific biases [33, 42] since it is a static method having the closest formulation to ours. We randomly initialize the embeddings and bias terms, and train the model with the Adam optimizer [35] for 500 epochs and a learning rate of 0.1. A very well-known GRL method, (ii) NODE2VEC [7] relies on the explicit generation of the random walks by starting from each node in the network, then it learns node embeddings by inspiring from the SkipGram [43] algorithm. It optimizes the softmax function for the nodes lying within a fixed window region with respect to a chosen center node over the produced node sequences. In our experiments, we tune the model's parameters $(p, q)$ from $\{0.25, 0.5, 1, 2, 4\}$. Since it has the ability to run over the weighted networks, we also constructed a weighted graph based on the number of links through time and reported the best score of both versions of the networks. (iii) CTDNE [24] is a dynamic node embedding approach performing temporal random walks over the network. (iv) HTNE [25] learns embeddings based on the Hawkes process modeling the neighborhood formation sequence induced from the network structure. (v) MMDNE [26] introduces a temporal attention point process to model the newly established links and proposes a general dynamics equation relying on latent node representations to capture the network scale evolutions.

The continuous-time baseline methods are unable to produce instantaneous node representations and they produce embeddings only for a given time. Therefore, we have utilized the last time of the training set to obtain the representations. We have chosen the recommended values for the common hyper-parameters of NODE2VEC and CTDNE, so the number of walks, walk length, and window size parameters have been set to 10, 80, and 10, respectively. We used the implementation provided by the StellarGraph Python package to produce the embeddings for CTDNE. Similarly, we have adapted the suggested hyperparameter settings for MMDNE and CTDNE with 100 epochs.

For our method, we set the parameter $K = 25$, and bins count $B = 100$ to have enough capacity to track node interactions. For the regularization term ($\lambda$) of the prior, we first mask 20% of the dyads in the optimization of Equation (5). Furthermore, we train the model by starting with $\lambda = 10^6$, and then we reduce it to one-tenth after each 100 epoch. The same procedure is repeated until $\lambda = 10^{-6}$, and we choose the $\lambda$ value minimizing the log-likelihood of the masked pairs. The final embeddings are then obtained by performing this annealing strategy without any mask until this $\lambda$ value. We repeat this procedure 5 times, and we consider the best-performing method in learning the embeddings. The Coefficient of Variation (CV) of the experiments is always less than 0.5, and Figure 2a shows an illustrative example for tuning $\lambda$ over the *Synthetic($\pi$)* dataset with 5 random runs.

For the performance comparison of the methods, we provide the Area Under Curve (AUC) scores for the Receiver Operating Characteristic (ROC) and Precision-Recall (PR) curves [44]. We compute the intensity of a given instance for LDM and PIVEM for the similarity measure of the node pair. Since NODE2VEC and CTDNE rely on the SkipGram architecture [43], we use cosine similarity for them.

**Table 3:** The performance evaluation for the link prediction experiment over various datasets.

| | Synthetic($\pi$) | | Synthetic($\mu$) | | College | | Contacts | | Email | | Forum | | Hypertext | |
|---|---|---|---|---|---|---|---|---|---|---|---|---|---|---|
| | ROC | PR | ROC | PR | ROC | PR | ROC | PR | ROC | PR | ROC | PR | ROC | PR |
| LDM | .562 | .539 | .498 | **.642** | **.951** | **.944** | .860 | .835 | .954 | .948 | **.909** | **.897** | **.819** | .797 |
| NODE2VEC | .518 | .506 | .498 | .502 | .705 | .676 | .783 | .716 | .825 | .807 | .635 | .605 | .748 | .739 |
| CTDNE | .680 | .629 | .481 | .487 | .691 | .711 | .842 | .815 | .824 | .815 | .664 | .642 | .699 | .734 |
| HTNE | .573 | .569 | .491 | .493 | .715 | .684 | .864 | .824 | .838 | .837 | .764 | .747 | .785 | **.820** |
| MMDNE | .591 | .575 | **.506** | .515 | .717 | .703 | .874 | .847 | .827 | .832 | .762 | .746 | .795 | .813 |
| PIVEM | **.716** | **.689** | .474 | .485 | .891 | .887 | **.876** | **.884** | **.964** | **.964** | .894 | .895 | .756 | .767 |

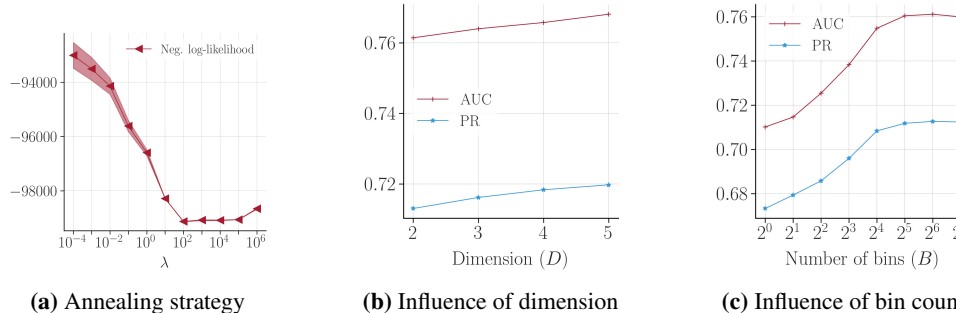

**(a)** Annealing strategy    **(b)** Influence of dimension    **(c)** Influence of bin count

**Figure 2:** Influence of the model hyperparameters over the *Synthetic($\pi$)* dataset.

**Network Reconstruction.** Our goal is to see how accurately a model can capture the interaction patterns among nodes and generate embeddings exhibiting their temporal relationships in a latent space. In this regard, we train the models on the residual network and generate sample sets as described previously. The performance of the models is reported in Table 1. Comparing the performance of PIVEM against the baselines, we observe favorable results across all networks, highlighting the importance and ability of PIVEM to account for and detect structure in a continuous time manner.

**Network Completion.** The network completion experiment is a relatively more challenging task than the reconstruction. Since we hide $10\%$ of the network, the dyads containing events are also viewed as non-link pairs, and the temporal models should place these nodes in distant locations of the embedding space. However, it might be possible to predict these events accurately if the network links have temporal triangle patterns through certain time intervals. In Table 2, we report the AUC-ROC and PR-AUC scores for the network completion experiment. Once more, PIVEM outperforms the baselines (in most cases significantly). We again discovered evidence supporting the necessity for modeling and tracking temporal networks with time-evolving embedding representations.

**Future Prediction.** Finally, we examine the performance of the models in the future prediction task. Here, the models are asked to forecast the $10\%$ future of the timeline. For PIVEM, the similarity between nodes is obtained by calculating the intensity function for the timeline of the training set (i.e., from 0 to 0.9), and we keep our previously described strategies for the baselines since they generate the embeddings only for the last training time. Table 3 presents the performances of the models. It is noteworthy that while PIVEM outperforms the baselines significantly on the *Synthetic($\pi$)* network, it does not show promising results on *Synthetic($\mu$)*. Since the first network is compatible with our model, it successfully learns the dominant link pattern of the network. However, the second network conflicts with our model: it forms a completely different structure for every $0.1$ second. For the real datasets, we observe mostly on-par results, especially with LDM. Some real networks contain link patterns that become "static" with respect to the future prediction task.

We have previously described how we set the prior coefficient, $\lambda$, and now we will examine the influence of the other hyperparameters over the *Synthetic($\pi$)* dataset for network reconstruction.

**Influence of dimension size ($D$).** We report the AUC-ROC and AUC-PR scores in Figure 2b. When we increase the dimension size, we observe a constant increase in performance. It is not a surprising result because we also increase the model's capacity depending on the dimension. However, the two-dimensional space still provides comparable performances in the experiments, facilitating human insights into networks' complex, evolving structures.

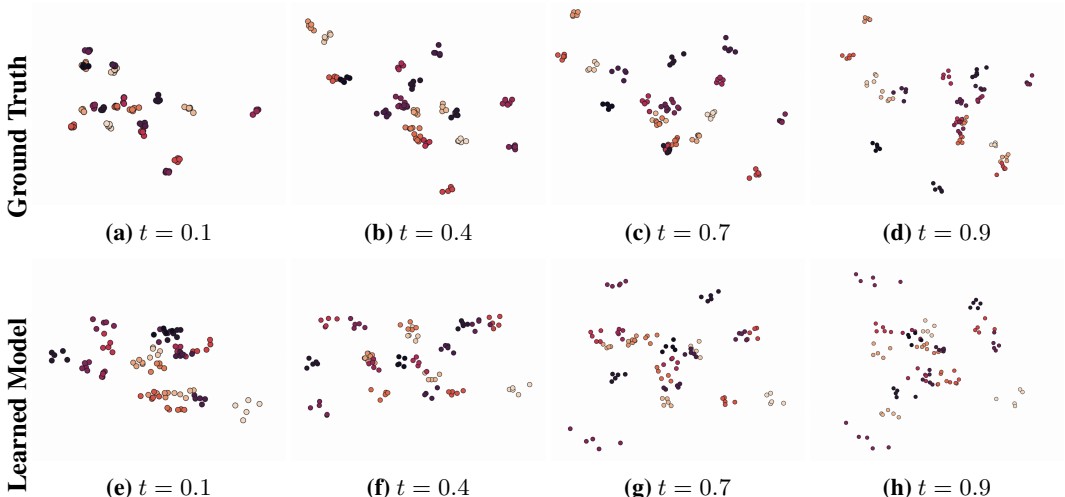

**Figure 3:** Comparisons of the ground truth and learned representations in two-dimensional space.

**Influence of bin count ($B$).** Figure 2c demonstrates the effect of the number of bins for the network reconstruction task. We generated the *Synthetic(π)* network from for $100$ bins, so it can be seen that the performance stabilizes around $2^6$, which points out that PIVEM reaches enough capability to model the interactions among nodes.

**Latent Embedding Animation.** It is of great significance from a human perspective to embed nodes into a latent space and visualize them in order to comprehend the intricate interactions among the entities and extract noteworthy information from the data. Although many GRL methods show high performance in the downstream tasks, in general, they require high dimensional spaces, so a postprocessing step later has to be applied in order to visualize the node representations in a small dimensional space. However, such processes cause distortions in the embeddings, which can lead a practitioner to end up with inaccurate arguments about the data.

As we have seen in the experimental evaluations, our proposed approach successfully learns embeddings in the two-dimensional space, and it also produces continuous-time representations. Therefore, it offers the ability to animate how the network evolves through time and can play a crucial role in grasping the underlying characteristics of the networks. As an illustrative example, Figure 3 compares the ground truth representations of *Synthetic(π)* with the learned ones. The synthetic network consists of small communities of 5 nodes, and each color indicates these groups. Although the problem does not have unique solutions, it can be seen that our model successfully seizes the clustering patterns in the network. We refer the reader to supplementary materials for the full animation.

## 5    Conclusion and Limitations

In this paper, we have proposed a novel continuous-time dynamic network embedding approach, namely, Piecewise Velocity Model (PIVEM). Its performance has been examined in various experiments, such as network reconstruction and completion tasks over various networks with respect to the very well-known baselines. We demonstrated that it could accurately embed the nodes into a two-dimensional space. Therefore, it can be directly utilized to animate the learned node embeddings, and it can be beneficial in extracting the networks' underlying characteristics, foreseeing how they will evolve through time. We theoretically showed that the model could scale up to large networks.

Although our model successfully learns continuous-time representations, it is unable to capture temporal patterns in the network in terms of the GP structure. Therefore, we are planning to employ different kernels instead of RBF, such as periodic kernels in the prior. The optimization strategies of the proposed method might be improved to escape from local minima. As a possible future direction, the algorithm can also be adapted for other graph types, such as directed and multi-layer networks.

## Acknowledgements

We would like to thank the reviewers for the constructive feedback and their insightful comments. We would also like to thank Sune Lehmann, Louis Boucherie, Lasse Mohr Mikkelsen, Simon Tommerup, August Semrau Andersen and William Diedrichsen Marstrand for the valuable and fruitful discussions. We gratefully acknowledge the Independent Research Fund Denmark for supporting this work [grant number: 0136-00315B].

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

# A  Appendix

In the main paper, we could not clarify every aspect of the model due to the restrictions on the number of pages. Hence, we will provide more detailed explanations here about the experiments, the computational problems, the proofs of the theoretical arguments, and possible extensions of the model toward the bipartite, weighted, and directed networks.

## A.1  Experiments

We consider all networks used in the experiments as undirected, and the event times of links are scaled to the interval $[0, 1]$ for the consistency of experiments. We use the finest resolution level of the given input timestamps, such as seconds and milliseconds. We provide a brief summary of the networks below, and various statistics are reported in Table 4. The visualization of the event distributions of the networks through time is depicted in Figure 4.

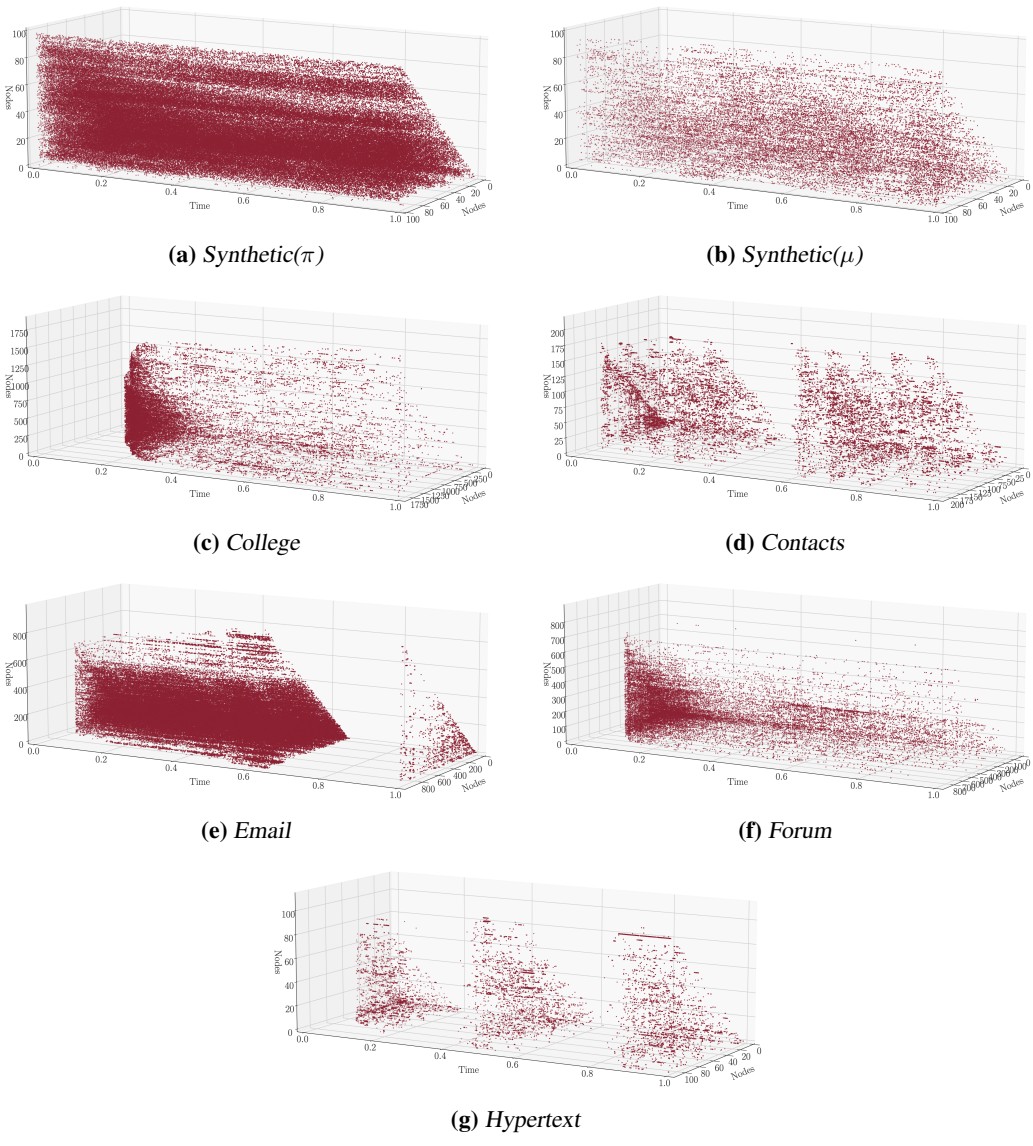

**(a)** *Synthetic($\pi$)*  **(b)** *Synthetic($\mu$)*

**(c)** *College*  **(d)** *Contacts*

**(e)** *Email*  **(f)** *Forum*

**(g)** *Hypertext*

**Figure 4:** Distribution of the links through time.

**Table 4:** Statistics of networks. $|\mathcal{V}|$: Number of nodes, $M$: Number of pairs having at least one link, $|\mathcal{E}|$: Total number of links, $\left.|\mathcal{E}_{ij}|\right|_{max}$: Max. number of links a pair of nodes has.

|  | $|\mathcal{V}|$ | $M$ | $|\mathcal{E}|$ | $\left.|\mathcal{E}_{ij}|\right|_{max}$ |
|---:|---|---|---|---|
| *Synthetic($\mu$)* | 100 | 4,889 | 180,658 | 124 |
| *Synthetic($\pi$)* | 100 | 3,009 | 22,477 | 32 |
| *College* | 1,899 | 13,838 | 59,835 | 184 |
| *Contacts* | 217 | 4,274 | 78,249 | 1,302 |
| *Hypertext* | 113 | 2,196 | 20,818 | 1,281 |
| *Email* | 986 | 16,064 | 332,334 | 4,992 |
| *Forum* | 899 | 7,036 | 33,686 | 171 |

**Synthetic datasets.** We generate two artificial networks in order to evaluate the behavior of the models in controlled experimental settings. **(i)** *Synthetic($\pi$)* is sampled from the prior distribution stated in Subsection 3.2. The hyper-parameters, $\boldsymbol{\beta}$, $K$ and $B$ are set to $\mathbf{0}$, 20 and 100, respectively. **(ii)** *Synthetic($\mu$)* is constructed based on the temporal block structures. The timeline is divided into 10 intervals, and the node set is split into 20 groups. The links within each group are sampled from the Poisson distribution with the constant intensity of 5.

**Real Networks.** The **(iii)** *Hypertext* network [37] was built on the radio badge records showing the interactions of the conference attendees for 2.5 days, and each event time indicates 20 seconds of active contact. Similarly, **(iv)** the *Contacts* network [38] was generated concerning the interactions of the individuals in an office environment. **(v)** *Forum* [39] is comprised of the activity data of university students on an online social forum system. The **(vi)** *CollegeMsg* network [40] indicates the private messages among the students on an online social platform. Finally, **(vii)** *Eu-Email* [41] was constructed based on the exchanged e-mail information among the members of European research institutions.

## A.2 Computational Problems, Model Complexity and Optimization Strategy

**Log-likelihood function.** Note that we need to evaluate the log-intensity term in Equation 5 for each $(i,j) \in \mathcal{V}^2$ $(i < j)$ and event time $e_{ij} \in \mathcal{E}_{ij}$. Therefore, the computational cost required for the whole network is bounded by $\mathcal{O}\left(|\mathcal{V}|^2|\mathcal{E}|\right)$. However, we can alleviate it by computing certain coefficients at the beginning of the optimization process. If we define $\alpha_{ij} := (e_{ij} - \Delta_B(b^* - 1))$, then it can be seen that the sum over the set of all events, $\mathcal{E}_{ij}^{b^*}$, lying inside $b^*$'th bin (i.e., the events in $[\Delta_B(b^* - 1), \ \Delta_B b^*)$ can be rewritten by:

$$
\sum_{e_{ij} \in \mathcal{E}_{ij}^{b^*}} \log \lambda_{ij}(e_{ij}) = \sum_{e_{ij} \in \mathcal{E}_{ij}} \left( \beta_i + \beta_j - ||\mathbf{r}_i(e_{ij}) - \mathbf{r}_j(e_{ij})||^2 \right)
$$

$$
= \sum_{e_{ij} \in \mathcal{E}_{ij}^{b^*}} \left( \beta_i + \beta_j \right) + \sum_{e_{ij} \in \mathcal{E}_{ij}} \left|\left| \Delta\mathbf{x}_{ij}^{(0)} + \Delta_B \sum_{b=1}^{b^*-1} \Delta\mathbf{v}_{ij}^{(b)} + \Delta\mathbf{v}_{ij}^{(b^*)}(e_{ij} - \Delta_B(b^*-1)) \right|\right|^2
$$

$$
= \sum_{e_{ij} \in \mathcal{E}_{ij}^{b^*}} \left( \beta_i + \beta_j \right) + \sum_{e_{ij} \in \mathcal{E}_{ij}} \left( \alpha_{ij}^2 \left|\left| \Delta\mathbf{v}_{ij}^{(b^*)} \right|\right|^2 + \left( \Delta\mathbf{x}_{ij}^{(0)} + \Delta_B \sum_{b=1}^{b^*-1} \Delta\mathbf{v}_{ij}^{(b)} \right)^2 \right.
$$

$$
\left. + 2\alpha_{ij} \left\langle \Delta\mathbf{x}_{ij}^{(0)} + \Delta_B \sum_{b=1}^{b^*-1} \Delta\mathbf{v}_{ij}^{(b)}, \Delta\mathbf{v}_{ij}^{(b^*)} \right\rangle \right)
$$

$$
= \left| \mathcal{E}_{ij}^{b^*} \right| (\beta_i + \beta_j) + \alpha_2 \left|\left| \Delta\mathbf{v}_{ij}^{(b^*)} \right|\right|^2 + \sum_{e_{ij} \in \mathcal{E}_{ij}} \left( \Delta\mathbf{x}_{ij}^{(0)} + \Delta_B \sum_{b=1}^{b^*-1} \Delta\mathbf{v}_{ij}^{(b)} \right)^2
$$

$$
+ 2\alpha_1 \left\langle \Delta\mathbf{x}_{ij}^{(0)} + \Delta_B \sum_{b=1}^{b^*-1} \Delta\mathbf{v}_{ij}^{(b)}, \Delta\mathbf{v}_{ij}^{(b^*)} \right\rangle
$$

where $\alpha_1^{(b^*)} := \sum_{e_{ij} \in \mathcal{E}_{ij}} \alpha_{ij}$ and $\alpha_2^{(b^*)} := \sum_{e_{ij} \in \mathcal{E}_{ij}} \alpha_{ij}^2$. We can follow the same strategy for each bin, then the computational complexity can be reduced to $\mathcal{O}\left(|\mathcal{V}|^2 B\right)$

Since we use the squared Euclidean distance in the integral term of our objective, we can derive the exact formula for the computation (please see Lemma A.4 for the details). We need to evaluate it for all node pairs, so it requires at most $\mathcal{O}\left(|\mathcal{V}|^2\right)$ operations. Hence, the complexity of the log-likelihood function is $\mathcal{O}\left(|\mathcal{V}|^2 B\right)$. Instead of optimizing the whole network at once, we are applying the batching strategy over the set of nodes in order to reduce the memory requirements, so we sample $\mathcal{S}$ nodes for each epoch. Hence, the overall complexity of the log-likelihood is $\mathcal{O}\left(\mathcal{S}^2 B \mathcal{I}\right)$ where $\mathcal{I}$ is the number of epochs.

**Computation of the prior function.** The covariance matrix, $\boldsymbol{\Sigma} \in \mathbb{R}^{BND \times BND}$, of the prior is defined by $\boldsymbol{\Sigma} := \lambda^2 \left(\sigma_{\boldsymbol{\Sigma}}^2 \mathbf{I} + \mathbf{K}\right)^{-1}$ with a scaling factor $\lambda \in \mathbb{R}$ and a noise variance $\sigma_{\boldsymbol{\Sigma}}^2 \in \mathbb{R}^+$. The multivariate normal distribution is parametrized with a noise term $\sigma_{\boldsymbol{\Sigma}}^2 \mathbf{I}$ and a matrix $\mathbf{K} \in \mathbb{R}^{BND \times BND}$ having a low-rank form. In other words, $\mathbf{K}$ is written by $\mathbf{B} \otimes \mathbf{C} \otimes \mathbf{D}$ where $\mathbf{B}$ is block diagonal matrix combined with parameter $c_{\mathbf{x}^0}$ and the RBF kernel $\exp\left(-(c_b - c_{b'})^2 / \sigma_{\mathbf{B}}^2\right) \in \mathbb{R}^{B \times B}$ for $c_b := (t_{b-1} - t_b)/2$. The matrix aiming for capturing the node interactions, $\mathbf{C} := \mathbf{Q}\mathbf{Q}^\top \in \mathbb{R}^{N \times N}$ is defined with a low-rank matrix $\mathbf{Q} \in \mathbb{R}^{N \times k}$ whose rows equal to $1$ ($k \ll N$), and we set $\mathbf{D} := \mathbf{I}\,\mathbf{I}^\top \in \mathbb{R}^{D \times D}$. By considering the Cholesky decomposition [36] of $\mathbf{B} := \mathbf{L}\mathbf{L}^\top$ since $\mathbf{B}$ is symmetric positive semi-definite, we can factorize $\mathbf{K} := \mathbf{K}_f \mathbf{K}_f^\top$ where $\mathbf{K}_f := \mathbf{L} \otimes \mathbf{Q} \otimes \mathbf{I}$.

Note that the precision matrix, $\boldsymbol{\Sigma}^{-1}$, can be written by using the *Woodbury matrix identity* [36] as follows:

$$\boldsymbol{\Sigma}^{-1} = \lambda^{-2} \left(\sigma_{\boldsymbol{\Sigma}}^2 \mathbf{I} + \mathbf{K}_f \mathbf{K}_f^\top\right)^{-1} = \lambda^{-2} \left(\sigma_{\boldsymbol{\Sigma}}^{2^{-1}} \mathbf{I} - \sigma_{\boldsymbol{\Sigma}}^{2^{-1}} \mathbf{K}_f \mathbf{R}^{-1} \mathbf{K}_f^\top \sigma_{\boldsymbol{\Sigma}}^{2^{-1}}\right)$$

where the capacitance matrix $\mathbf{R} := \mathbf{I}_{BKD} + \sigma_{\boldsymbol{\Sigma}}^{2^{-1}} \mathbf{K}_f^\top \mathbf{K}_f$.

The log-determinant of $\lambda^2 \boldsymbol{\Sigma}$ can be also simplified by applying *Matrix Determinant lemma* [36]:

$$\begin{aligned}
\log(det(\boldsymbol{\Sigma})) &= (BND)\log\left(\lambda^2\right) + \log\left(det(\sigma_{\boldsymbol{\Sigma}}^2 \mathbf{I}_{BND} + \mathbf{K}_f \mathbf{K}_f^\top)\right) \\
&= (BND)\log\left(\lambda^2\right) + \log\left(det(\mathbf{I}_{BKD} + \sigma_{\boldsymbol{\Sigma}}^{2^{-1}} \mathbf{K}_f^\top \mathbf{K}_f)\right) + (BND)\log\left(\sigma_{\boldsymbol{\Sigma}}^2\right) \\
&= (BND)\left(\log(\lambda^2) + \log\left(\sigma_{\boldsymbol{\Sigma}}^2\right)\right) + \log(det(\mathbf{R}))
\end{aligned}$$

Note that the most cumbersome points in the computation of the prior are the calculations of the inverse and determinant of the terms and some matrix multiplication operations. Since $R$ is a matrix of size $BKD \times BKD$, its inverse and determinant can be found in at most $\mathcal{O}(B^3 K^3 D^3)$ operations. We also need the term, $\mathbf{K}_f \mathbf{R}^{-1} \mathbf{R}$, which can also be computed in $\mathcal{O}(B^3 D^3 K^2 |\mathcal{V}|)$ steps, so the number of operations required for the prior can be bounded by $\mathcal{O}(B^3 D^3 K^2 |\mathcal{V}|)$. It is worth noticing that we cannot directly apply the batching strategy for the computation of the inverse of the capacitance matrix, $\mathbf{R}$. However, we can compute it once and then we can utilize it for the calculation of the log-prior for different sets of node samples, then we can recompute it when we decide to update the parameters again. To sum up, the complexity of our proposed approach is $\mathcal{O}(B\mathcal{I}\mathcal{S}^2 + B^3 D^3 K^2 \mathcal{S}\mathcal{I})$ where $\mathcal{S}$ is the batch size and $\mathcal{I}$ is the number of epochs.

**Optimization of the proposed approach**. Our objective given in Equation (5) is not a convex function, thus the learning strategy that we follow is of great importance in order to escape from local minima of poor quality representations. We start by randomly initializing the model's hyper-parameters from $[-1, 1]$ except for the velocity tensor, which is set to $0$ at the beginning. We adapt a sequential learning strategy for the learning of these parameters. In other words, we first optimize the initial position and bias terms together, $\{\mathbf{x}^{(0)}, \boldsymbol{\beta}\}$, for 33 epochs; then, we include the velocity tensor, $\{\mathbf{v}\}$, into the optimization process and repeat the training for the same number of epochs. Finally, we add the prior parameters and learn all model hyper-parameters together. We have employed *Adam optimizer* [35] with a learning rate of $0.1$.

In our experiments, we set the parameter $K = 25$, and bins count $B = 100$ to have enough capacity to track node interactions. In order to find an optimal regularization term $\lambda$ value and to determine the influence of the prior in the objective, we apply an annealing strategy for the model. We first mask 20% of the dyads during the optimization of Equation (5). Furthermore, we train the model by

starting with $\lambda = 10^6$ and learn all parameters using the sequential optimization strategy. We then gradually reduce $\lambda$ to one-tenth upon optimizing all model parameters for 100 epochs. The same procedure is repeated until $\lambda = 10^{-6}$. We choose the $\lambda$ value minimizing the log-likelihood of the masked pairs (i.e., based on the predictive log-likelihood evaluated on these pairs).

The final node embeddings are then obtained by performing this annealing strategy without any mask until the found ideal $\lambda$ value. We repeat this procedure 5 times with different initializations, and we consider the best-performing method/seed value in learning the final embeddings. The relative standard deviation of the experiments is always less than 0.5 for all the networks, and we display the negative log-likelihood of the masked pairs for the annealing strategy with 5 random runs in Figure 5. The blue curves demonstrate the same annealing strategy but in the opposite order. In other words, we start from a very restrictive model with low $\lambda$ value and increase $\lambda$ to have a more flexible model.

The considered annealing strategy thereby quantifies the impact for different strengths of imposing the GP prior. It corresponds to a highly constrained model akin to static representations for small values of $\lambda$ in which the GP prior has close to zero variance of the parameters to highly flexible dynamic representations almost entirely driven by the likelihood function for high values of $\lambda$. The annealing strategy thus highlights the GP prior's impact and the optimal regime imposing such prior.

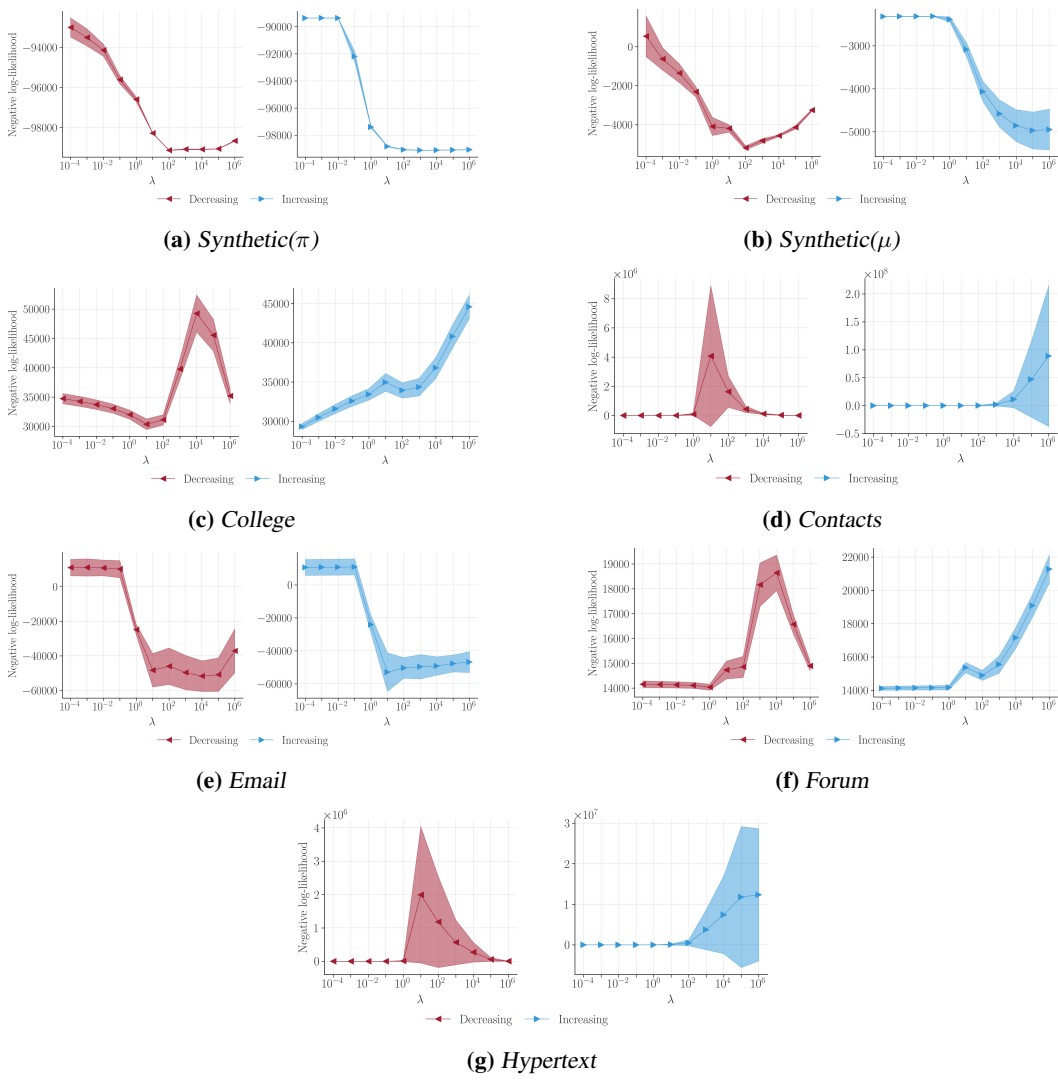

**Figure 5:** Negative log-likelihood of the masked pairs for the annealing strategy applied for tuning $\lambda$ parameter with 5 random runs.

### A.3 Theoretical Results

**Lemma A.1.** *For given fixed bias terms $\{\beta_i\}_{i\in\mathcal{V}}$, the node embeddings, $\{\mathbf{r}_i(t)\}_{i\in\mathcal{V}}$, learned by the objective given in Equation 1 within a bounded set of radius $R_t$ during a time interval $[t_l, t_u]$ satisfy*

$$\log\left(\frac{(t_u - t_l)}{-\log p_{ij}^0}\right) + (\beta_i + \beta_j) \leq \frac{1}{(t_u - t_l)}\int_{t_l}^{t_u}||\mathbf{r}_i(t) - \mathbf{r}_j(t)||^2 dt \leq \log\left(\frac{(t_u - t_l)}{-\log(1 - p_{ij}^>)}\right) + (\beta_i + \beta_j) + R_t$$

*where $p_{ij}^0$ and $p_{ij}^>$ are the probabilities of having zero and more than zero links for the nodes $i$ and $j$.*

*Proof.* Let $p_{ij}^0 := \mathbb{P}\{X_{ij} = 0\}$, and it can be written that

$$p_{ij}^0 = \exp(-\lambda_{ij}) = \exp\left(-\int_{t_l}^{t_u}\exp\left(\beta_i + \beta_j - ||\mathbf{r}_i(t) - \mathbf{r}_j(t)||^2\right)dt\right)$$

$$= \exp\left(-\exp(\beta_i + \beta_j)\int_{t_l}^{t_u}\exp\left(-||\mathbf{r}_i(t) - \mathbf{r}_j(t)||^2\right)dt\right)$$

Then, the lower bound can be derived by

$$\frac{-\log(p_{ij}^0)}{(t_u - t_l)}\frac{1}{\exp(\beta_i + \beta_j)} = \frac{1}{(t_u - t_l)}\int_{t_l}^{t_u}\exp\left(-||\mathbf{r}_i(t) - \mathbf{r}_j(t)||^2\right)dt$$

$$\geq \exp\left(-\frac{1}{(t_u - t_l)}\int_{t_l}^{t_u}||\mathbf{r}_i(t) - \mathbf{r}_j(t)||^2 dt\right)$$

where the second line follows from the Jensen's inequality, and it can be written that:

$$\frac{1}{(t_u - t_l)}\int_{t_l}^{t_u}||\mathbf{r}_i(t) - \mathbf{r}_j(t)||^2 dt \geq \log\left(\frac{(t_u - t_l)}{-\log p_{ij}^0}\right) + (\beta_i + \beta_j).$$

Based on a similar strategy that we have applied for the lower bound, we can write

$$\frac{-\log(1 - p_{ij}^>)}{(t_u - t_l)}\frac{1}{\exp(\beta_i + \beta_j)} = \frac{1}{(t_u - t_l)}\int_{t_l}^{t_u}\exp\left(-||\mathbf{r}_i(t) - \mathbf{r}_j(t)||^2\right)dt$$

$$\leq \exp\left(R_t\right)\exp\left(-\frac{1}{(t_u - t_l)}\int_{t_l}^{t_u}||\mathbf{r}_i(t) - \mathbf{r}_j(t)||^2 dt\right)$$

where the inequality follows from Lemma A.2 in which $f(t)$ corresponds to $||\mathbf{r}_i(t) - \mathbf{r}_j(t)||^2$ and $R_t := \sup_{t\in[t_l,t_u]}||\mathbf{r}_i(t) - \mathbf{r}_j(t)||^2$. Then, it can be concluded that

$$\frac{1}{(t_u - t_l)}\int_{t_l}^{t_u}||\mathbf{r}_i(t) - \mathbf{r}_j(t)||^2 dt \leq \log\left(\exp(\beta_i + \beta_j)\exp\left(R_t\right)\frac{(t_u - t_l)}{-\log(1 - p_{ij}^>)}\right)$$

$$\leq (\beta_i + \beta_j) + R_t + \log\left(\frac{(t_u - t_l)}{-\log(1 - p_{ij}^>)}\right)$$

$\square$

**Lemma A.2.** *Let $f : [t_l, t_u] \to \mathbb{R}^+$ be a real-valued non-negative continuous function, then it satisfies*

$$\frac{1}{(t_u - t_l)} \int_{t_l}^{t_u} \exp(-f(t)) dt \leq \exp(R_t) \exp\left(-\frac{1}{(t_u - t_l)} \int_{t_l}^{t_u} f(t) dt\right)$$

*for some constant $R_t := \sup_{t \in [t_l, t_u]} f(t) \in \mathbb{R}$.*

*Proof.* Let $\hat{f} := \frac{1}{(t_u - t_l)} \int_{t_l}^{t_u} f(t) dt$ for some $t_u, t_l \in \mathbb{R}$ where $t_l < t_u$, then it can be written that

$$\frac{1}{(t_u - t_l)} \int_{t_l}^{t_u} \exp(-f(t) + \hat{f}) dt \leq \frac{1}{(t_u - t_l)} \int_{t_l}^{t_u} \exp(\hat{f}) dt$$

$$= \frac{1}{(t_u - t_l)} \int_{t_l}^{t_u} \exp\left(\frac{1}{(t_u - t_l)} \int_{t_l}^{t_u} f(t') dt'\right) dt$$

$$\leq \frac{1}{(t_u - t_l)} \int_{t_l}^{t_u} \exp\left(\sup_{t^* \in [t_l, t_u]} f(t^*)\right) dt$$

$$= \exp(R_t)$$

where $R_t := \sup_{t \in [t_l, t_u]} f(t)$, and we have the first inequality since $f(t)$ is a non-negative function. In conclusion, we can obtain

$$\frac{1}{(t_u - t_l)} \int_{t_l}^{t_u} \exp(-f(t)) dt \leq \exp(R_t) \exp\left(-\frac{1}{(t_u - t_l)} \int_{t_l}^{t_u} f(t) dt\right)$$

$\square$

**Theorem A.3.** *Let $\mathbf{f}(t) : [0, T] \to \mathbb{R}^D$ be a continuous embedding of a node. For any given $\epsilon > 0$, there exists a continuous, piecewise-linear node embedding, $\mathbf{r}(t)$, satisfying $||\mathbf{f}(t) - \mathbf{r}(t)||_2 < \epsilon$ for all $t \in [0, T]$ where $\mathbf{r}(t) := \mathbf{r}^{(b)}(t)$ for all $(b-1)\Delta_B \leq t < b\Delta_B$, $\mathbf{r}(t) := \mathbf{r}^{(B)}(t)$ for $t = T$ and $\Delta_B = T/B$ for some $B \in \mathbb{N}^+$.*

*Proof.* Let $\mathbf{f}(t) : [0, T] \to \mathbb{R}^D$ be a continuous embedding so it is also uniformly continuous by the Heine–Cantor theorem since $[0, T]$ is a compact set. Then, we can find some $B \in \mathbb{N}^+$ such that for every $t, \tilde{t} \in [0, T]$ with $|t - \tilde{t}| \leq \Delta_B := T/B$ implies $||\mathbf{f}(t) - \mathbf{f}(\tilde{t})||_2 < \epsilon/2$ for any given $\epsilon > 0$.

Let us define $\mathbf{r}^{(b)}(t) = \mathbf{r}^{(b-1)}\left((b-1)\Delta_B\right) + \mathbf{v}_b(t - (b-1)\Delta_B)$ recursively for each $b \in \{1, \ldots, B\}$ where $\mathbf{r}^{(0)}(0) := \mathbf{x}_0 = \mathbf{f}(0)$, and $\mathbf{v}_b := \frac{\mathbf{f}(b\Delta_B) - \mathbf{f}\left((b-1)\Delta_B\right)}{\Delta_B}$. Then it can be seen that we have $\mathbf{r}^{(b)}(b\Delta_B) = \mathbf{f}(b\Delta_B)$ for all $b \in \{1, \ldots, B\}$ because

$$\mathbf{r}^{(b)}(b\Delta_B) = \mathbf{r}^{(b-1)}\left((b-1)\Delta_B\right) + \mathbf{v}_b\left(b\Delta_B - \Delta_B(b-1)\right)$$

$$= \mathbf{r}^{(b-1)}\left((b-1)\Delta_B\right) + \mathbf{v}_b\Delta_B$$

$$= \mathbf{r}^{(b-1)}\left((b-1)\Delta_B\right) + \left(\frac{\mathbf{f}(b\Delta_B) - \mathbf{f}\left((b-1)\Delta_B\right)}{\Delta_B}\right)\Delta_B$$

$$= \mathbf{r}^{(b-1)}\left((b-1)\Delta_B\right) + \left(\mathbf{f}(b\Delta_B) - \mathbf{f}\left((b-1)\Delta_B\right)\right)$$

$$= \mathbf{r}^{(b-2)}\left((b-2)\Delta_B\right) + \left(\mathbf{f}\left((b-1)\Delta_B\right) - \mathbf{f}((b-2)\Delta_B)\right) + \left(\mathbf{f}(b\Delta_B) - \mathbf{f}\left((b-1)\Delta_B\right)\right)$$

$$= \mathbf{r}^{(b-2)}\left((b-2)\Delta_B\right) + \left(\mathbf{f}(b\Delta_B) - \mathbf{f}((b-2)\Delta_B)\right)$$

$$= \cdots$$

$$= \mathbf{r}^{(0)}(0) + \left(\mathbf{f}(b\Delta_B) - \mathbf{f}(0)\right)$$

$$= \mathbf{f}(b\Delta_B)$$

where the last line follows from the fact that $\mathbf{r}^{(0)}(0) = \mathbf{x}_0 = \mathbf{f}(0)$ by the definition. Therefore, for any given point $t \in [0, T)$ for $b = \lfloor t/\Delta_b \rfloor + 1$, it can be seen that

$$
\begin{aligned}
||\mathbf{f}(t) - \mathbf{r}(t)||_2 &= ||\mathbf{f}(t) - \mathbf{r}^{(b)}(t)||_2 \\
&= \left|\left|\mathbf{f}(t) - \left(\mathbf{r}^{(b-1)}((b-1)\Delta_B) + \mathbf{v}_b(t - (b-1)\Delta_B)\right)\right|\right|_2 \\
&= \left|\left|\mathbf{f}(t) - \left(\mathbf{r}^{(b-1)}((b-1)\Delta_B) + \left(\frac{\mathbf{f}(b\Delta_B) - \mathbf{f}((b-1)\Delta_B)}{\Delta_B}\right)(t - (b-1)\Delta_B)\right)\right|\right|_2 \\
&= \left|\left|\left(\mathbf{f}(t) - \mathbf{r}^{(b-1)}((b-1)\Delta_B)\right) + \left(\mathbf{f}(b\Delta_B) - \mathbf{f}((b-1)\Delta_B)\right)\left(\frac{t - (b-1)\Delta_B}{\Delta_B}\right)\right|\right|_2 \\
&\leq \left|\left|\mathbf{f}(t) - \mathbf{r}^{(b-1)}((b-1)\Delta_B)\right|\right| + \left|\left|\mathbf{f}(b\Delta_B) - \mathbf{f}((b-1)\Delta_B)\right|\right| \\
&< \frac{\epsilon}{2} + \frac{\epsilon}{2} \\
&= \epsilon
\end{aligned}
$$

where the inequality in the fifth line holds since we have $\left|\frac{t-(b-1)\Delta_B}{\Delta_B}\right| \leq 1$ $\qquad\square$

**Lemma A.4** (Integral Computation). *The integral of the intensity function, $\lambda_{ij}(t)$, from $t_l$ to $t_u$ is equal to*

$$
\int_{t_l}^{t_u} \exp\left(\beta_{ij} - \left\|\Delta\mathbf{x}_{ij} + \Delta\mathbf{v}_{ij}t\right\|^2\right) = \frac{\sqrt{\pi}\exp\left(\beta_{ij} + r_{ij}^2 - \left\|\Delta\mathbf{x}_{ij}\right\|^2\right)}{2\left\|\Delta\mathbf{v}_{ij}\right\|}\mathrm{erf}\left(\left\|\Delta\mathbf{v}_{ij}\right\| t + r_{ij}\right)\Bigg|_{t=t_l}^{t=t_u}
$$

*where $\beta_{ij} := \beta_i + \beta_j$, $\Delta\mathbf{x}_{ij} := \mathbf{x}_i^{(0)} - \mathbf{x}_j^{(0)}$, $\Delta\mathbf{v}_{ij} := \mathbf{v}_i^{(1)} - \mathbf{v}_j^{(1)}$ and $r := \frac{\langle\Delta\mathbf{v}_{ij}, \Delta\mathbf{x}_{ij}\rangle}{\left\|\Delta\mathbf{v}_{ij}\right\|}$.*

*Proof.*

$$
\begin{aligned}
\int_{t_l}^{t_u} \exp\left(-\left\|\Delta\mathbf{x}_{ij} + \Delta\mathbf{v}_{ij}t\right\|^2\right) &= \int_{t_l}^{t_u} \exp\left(-\left\|\Delta\mathbf{v}_{ij}\right\|^2 t^2 - 2\langle\Delta\mathbf{x}_{ij}, \Delta\mathbf{v}_{ij}\rangle t - \left\|\Delta\mathbf{x}_{ij}\right\|^2\right)\mathrm{d}t \\
&= \int_{t_l}^{t_u} \exp\left(-\left(\left\|\Delta\mathbf{v}_{ij}\right\| t + r_{ij}\right)^2 + r_{ij}^2 - \left\|\Delta\mathbf{x}_{ij}\right\|^2\right)\mathrm{d}t \qquad (6)
\end{aligned}
$$

where $r_{ij} := \frac{\langle\Delta\mathbf{v}_{ij}, \Delta\mathbf{x}_{ij}\rangle}{\left\|\Delta\mathbf{v}_{ij}\right\|}$. The substitution $u = \left\|\Delta\mathbf{v}_{ij}\right\| t + r_{ij}$ yields $\mathrm{d}u = \left\|\Delta\mathbf{v}_{ij}\right\| \mathrm{d}t$. Furthermore, we have

$$
\begin{aligned}
\int_{t_l}^{t_u} \exp\left(-\left(\left\|\Delta\mathbf{v}_{ij}\right\| t + r_{ij}\right)^2\right)\mathrm{d}t &= \frac{1}{\left\|\Delta\mathbf{v}_{ij}\right\|} \int_{\left\|\Delta\mathbf{v}_{ij}\right\| t_l + r_{ij}}^{\left\|\Delta\mathbf{v}_{ij}\right\| t_u + r_{ij}} \exp\left(-u^2\right)\mathrm{d}u \\
&= \frac{1}{\left\|\Delta\mathbf{v}_{ij}\right\|}\frac{\sqrt{\pi}}{2}\left(\frac{2}{\sqrt{\pi}} \int_{\left\|\Delta\mathbf{v}_{ij}\right\| t_l + r_{ij}}^{\left\|\Delta\mathbf{v}_{ij}\right\| t_u + r_{ij}} \exp\left(-u^2\right)\mathrm{d}u\right) \\
&= \frac{\sqrt{\pi}}{2\left\|\Delta\mathbf{v}_{ij}\right\|}\mathrm{erf}\left(\left\|\Delta\mathbf{v}_{ij}\right\| t + r_{ij}\right)\Bigg|_{t=t_l}^{t=t_u} \qquad (7)
\end{aligned}
$$

By using Equations 6 and 7, it can be obtained that

$$
\begin{aligned}
\int_{t_l}^{t_u} \exp\left(-\left\|\Delta\mathbf{x}_{ij} + \Delta\mathbf{v}_{ij}t\right\|^2\right) &= \exp\left(r_{ij}^2 - \left\|\Delta\mathbf{x}_{ij}\right\|^2\right)\int_{t_l}^{t_u} \exp\left(-\left(\left\|\Delta\mathbf{v}_{ij}\right\| t + r_{ij}\right)^2\right)\mathrm{d}t \\
&= \frac{\sqrt{\pi}\exp\left(r_{ij}^2 - \left\|\Delta\mathbf{x}_{ij}\right\|^2\right)}{2\left\|\Delta\mathbf{v}_{ij}\right\|}\left[\mathrm{erf}\left(\left\|\Delta\mathbf{v}_{ij}\right\| t + r_{ij}\right)\Bigg|_{t=t_l}^{t=t_u}\right.
\end{aligned}
$$

Therefore, we can conclude that

$$\int_{t_l}^{t_u} \exp\left(\beta_{ij} - \left\|\Delta\mathbf{x}_{ij} + \Delta\mathbf{v}_{ij}t\right\|^2\right) = \frac{\sqrt{\pi}\exp\left(\beta_{ij} + r_{ij}^2 - \left\|\Delta\mathbf{x}_{ij}\right\|^2\right)}{2\left\|\Delta\mathbf{v}_{ij}\right\|}\mathrm{erf}\left(\left\|\Delta\mathbf{v}_{ij}\right\|t + r_{ij}\right)\Bigg|_{t=t_l}^{t=t_u}$$

$\square$

### A.4  Extension to Weighted, Directed, and Bipartite Networks

In this section, we will discuss how the proposed approach, PIVEM, can be extended for weighted, directed, and bipartite networks.

**Weighted networks.** Our approach can be simply adapted for positive integer-weighted networks by replacing each weighted link in the network with multiple unit events corresponding to the integer weight at the specific time point of the integer-weighted link. Then, PIVEM can be run as is for the reinterpreted version of the network without making any modifications to the structure of the approach.

**Directed and Bipartite networks.** Let $\mathcal{G} = ((\mathcal{V}, \mathcal{U}), \mathcal{E})$ be a bipartite network with the parts $\mathcal{V} = \{v_1, \ldots, v_{N_1}\}$ and $\mathcal{U} = \{u_1, \ldots, u_{N_2}\}$. We can rewrite the objective given in Equation 5 by considering only the pairs $(i, j) \in \mathcal{V} \times \mathcal{U}$ belonging to different parts:

$$\hat{\Omega} = \arg\max_{\Omega} \sum_{i \in \mathcal{V}} \sum_{j \in \mathcal{U}} \left( \sum_{e_{ij} \in \mathcal{E}_{ij}} \log \lambda_{ij}(e_{ij}) - \int_0^T \lambda_{ij}(t)dt \right) + \log\mathcal{N}\left(\begin{bmatrix} \mathbf{x}^{(0)} \\ \mathbf{v} \end{bmatrix}; \mathbf{0}, \mathbf{\Sigma}\right) \quad (8)$$

where the intensity function, $\lambda_{ij}(e_{ij})$ is defined as follows:

$$\lambda_{ij}(t) := \exp\left(\beta_i + \eta_j - ||\mathbf{r}_i^*(t) - \mathbf{r}_j^{**}(t)||^2\right), \quad (9)$$

where $\beta_i$ and $\eta_j$ indicate the bias/random effect terms for the nodes belonging respectively to $\mathcal{V}$ an $\mathcal{U}$. Similarly, we can introduce distinct initial position $\mathbf{x}_i^*$, $\mathbf{x}_j^{**}$ and velocity tensors $\mathbf{v}_i^*$, $\mathbf{v}_j^{**}$ to define the node representations, $\mathbf{r}_i^*(t) \in \mathbb{R}^D$ and $\mathbf{r}_j^{**}(t) \in \mathbb{R}^D$ at time $t$. Note that we can write $\mathbf{x}_i^{(0)} = \mathbf{x}_i^* \oplus \mathbf{x}_i^{**}$, and $\mathbf{v}_i = \mathbf{v}_i^* \oplus \mathbf{v}_i^{**}$ where $\oplus$ indicates the tensor concatenation operation.

For the directed case, we specify the model similar to the bipartite case but define the likelihood function as

$$\hat{\Omega} = \arg\max_{\Omega} \sum_{i \neq j} \left( \sum_{e_{ij} \in \mathcal{E}_{ij}} \log \lambda_{ij}(e_{ij}) - \int_0^T \lambda_{ij}(t)dt \right) + \log\mathcal{N}\left(\begin{bmatrix} \mathbf{x}^{(0)} \\ \mathbf{v} \end{bmatrix}; \mathbf{0}, \mathbf{\Sigma}\right). \quad (10)$$

## A.5 Table of Symbols

The detailed list of the symbols used throughout the manuscript and their corresponding definitions can be found in Table 5.

**Table 5:** Table of symbols

| Symbol | Description |
|---:|---|
| $\mathcal{G}$ | Graph |
| $\mathcal{V}$ | Vertex set |
| $\mathcal{E}$ | Edge set |
| $\mathcal{E}_{ij}$ | Edge set of node pair $(i, j)$ |
| $N$ | Number of nodes |
| $D$ | Dimension size |
| $\mathcal{I}_T$ | Time interval |
| $T$ | Time length |
| $B$ | Number of bins |
| $\beta_i$ | Bias term of node $i$ |
| $\mathbf{x}$ | Initial position matrix |
| $\mathbf{v}^{(b)}$ | Velocity matrix for bin $b$ |
| $\mathbf{r}_i(t)$ | Position of node $i$ at time $t$ |
| $\lambda_{ij}(t)$ | Intensity of node pair $(i, j)$ at time $t$ |
| $e_{ij}$ | An event time of node pair $(i, j)$ |
| $\Sigma$ | Covariance matrix |
| $\lambda$ | Scaling factor of the covariance |
| $\sigma_{\mathbf{\Sigma}}$ | Noise variance |
| $\sigma_{\mathbf{B}}$ | Lengthscale variable of RBF kernel |
| $\otimes$ | Kronecker product |
| $\mathbf{I}$ | Identity matrix |
| $\mathbf{B}$ | Bin interaction matrix |
| $\mathbf{C}$ | Node interaction matrix |
| $\mathbf{D}$ | Dimension interaction matrix |
| $\mathbf{R}$ | Capacitance matrix |
| $K$ | Latent dimension of $\mathbf{C}$ |

