# OpenReview forum: "Piecewise-Velocity Model for Learning Continuous-time Dynamic Node Representations"
_logconference.io/LOG/2022/Conference — LoG 2022 Poster_

### Official Review · Reviewer_gUt1 · 2022-10-21

**Overall Score:** 8
**Confidence:** 4

**Review:**

### Overview

The paper deals with the problem of learning node embedding on time-varying graphs. It proposes an interesting model that combines a Poisson process to characterize the temporal networks dynamics and a velocity-based linear model to compute embeddings. The proposed model is empirically evaluated on both real and synthetic networks.

Overall the paper is well-written. Moreover, most of the arguments made are supported either by some theoretical or experimental analysis.


*Strengths:*
- The formulation of the objective function is very interesting and novel.
- Some of the claims are supported by rigorous theoretical analysis.
- Interesting experiments on both real and artificial networks.

*Limitations:*
-  The main limitation of the paper has to do with the choice of baseline models. A more detailed discussion follows below.

### Detailed comments

- What is the intuition behind the loss function of the model defined in eq. (2)? Besides, why a continues-time model is preferable compared to a discrete-time one?

- The paper claims that, as a consequence of Lemma 3.1, the homophily property is satisfied in the embedding space. How strong is the claim? How homophily is being defined in dynamic networks? These points should be further clarified in the paper.

- The idea of expressing the position of a node using a piecewise velocity model is very interesting. Nevertheless, in the final objective in eq. (5), after considering a Gaussian Process prior, the model ends up having quite a few hyperparameters. The paper discusses the complexity of the model. However, a more elaborate discussion about how to tune these hyperparameters is needed.

- The paper also contains a good amount of empirical analysis on both synthetic and real-world graphs. The authors have put emphasis to analyze the sensitivity of the model w.r.t. the different hyperparameters. Still, it is not evident to me how the  Gaussian Process prior contributes to the performance by smoothening  the motion in the latent space. Some ablation study could be helpful here.

- Regarding the baselines, three models have been used in the experiments. Despite being able to draw some meaningful conclusions, I would strongly encourage the authors to consider some more relevant baselines for dynamic networks. It is not clear to me why node2vec has been used here when other models for dynamic networks have been introduced. This part of the analysis can further be enhanced. In the work of Kazemi et al., “Representation Learning for Dynamic Graphs: A Survey”, JMLR 2020, several such models are discussed.


Typos:

- Line 147: missing .
- Line 164: node*s*

---

### Official Review · Reviewer_HCDz · 2022-10-22

**Overall Score:** 5
**Confidence:** 4

**Review:**

Summary:

This paper proposes a model to generate low dimensional (2 or 3 dimensions) network embeddings to visualize and characterize dynamic networks so that the evolving dynamics can be better understood. In the proposed model, the temporal node evolution of nodes is approximated by piecewise linear interpolations. A theoretical analysis of the proposed method is provided. Experiments on both synthetic and real-world data are conducted.

Strengths:

1. The motivation to provide better visualization of dynamic graphs is clear and reasonable.

2. Theoretical analysis is provided from different perspectives.

3. Experiments are also conducted from different perspectives and the proposed method outperforms the baselines significantly.

Weakness and questions:

1. In the Introduction, why is accounting the homophily essential to dynamic graph visualization?

2. Other methods to obtain embeddings and visualization are also mentioned, but the comparison is unclear. The advantages of the proposed model compared to other models should be explained in more detail in the Introduction.

3. The idea of the proposed method is hard to understand. In the abstract, the high level idea of the model design is not clearly explained, which is not provided in the Introduction or Method either. To help readers better understand the model, a clear explanation of the high-level design with essential details should be delivered in either the Introduction or the Method part.

4. Besides the link prediction and network reconstruction, is there any method that can directly compare the visualization performance of the proposed method and the baselines?

5. What is the network characterization? Is this investigated either theoretically or experimentally?


Overall, the main problem is that the method lacks clarity. Therefore, the current version is not sufficient to be accepted.

---

### Official Review · Reviewer_i1a8 · 2022-10-22

**Overall Score:** 8
**Confidence:** 4

**Review:**

This paper proposes the PIecewise-VElocity Model (PIVEM) for the representation learning of continuous-time dynamic networks. The method is novel and the results are solid. For the future prediction task, it is unclear why it forms a completely different structure for every 0.1 second. It is unclear how the method works on weighted and directed graphs.

---

### Official Review · Reviewer_qsic · 2022-10-22

**Overall Score:** 6
**Confidence:** 4

**Review:**

**Summary**
This paper proposes a continuous-time dynamic network embedding method named Piecewise Velocity Model (PIVEM). The model approximates temporal evolutions by piecewise linear interpolations based on a latent distance model with piecewise constant node-specific velocities. A structured Gaussian Process prior is proposed to further characterize the network dynamics. Experiments on 2 synthetic datasets and 5 real datasets in 3 downstream tasks (network reconstruction, network completion, and future prediction) show the superiority of the proposed method.

**Pros**
[+] The proposed method approximates the temporal evolution by piecewise linear interpolations with theoretical analysis (Theorem 3.2), which seems to be an interesting perspective.
[+] The authors provide a detailed time complexity analysis for the proposed method.
[+] Visualizations and animations of learned embeddings in the 2-D space help to understand the proposed method.
[+] The authors have provided the code to increase reproducibility.

**Cons and questions**
[-] From Eq. (4), it seems that v_i in each time bin is a learnable parameter, and needs to be optimized in the training stage. In this way, it seems that the proposed method can not tackle inductive settings, i.e. predict temporal evolution in the future as v_i in those time bins could not be trained in the training stage, which is a considerable limitation.
[-] The experiments are somewhat weak considering the following perspectives:
(1) The compared baselines should be enhanced. The authors only compare two static embedding methods and one dynamic method CTDNE from 2018. More recent SOTA baselines should be compared (which can be easily found in the provided survey or search engines).
(2) The authors mention that one advantage of the proposed method is scalability, while the adopted datasets are rather small-scale. Experiments on more large-scale datasets could further demonstrate the proposed method.
(3) The authors mention that the proposed method relies on the prior probability, i.e., Eq.(5). Some analysis could be performed to show whether the proposed method is sensitive to this prior.

Minor:
(1) Repeated reference, [2] and [10], [16] and [24], etc.
(2) Some writing could be improved.

---

### Meta-Review · Area_Chair_yGTx · 2022-11-21

**Confidence:** 5
**Recommendation:** Accept

**Meta Review:**

This paper introduces a continuous-time dynamic network embedding method named Piecewise Velocity Model. The consensus among PC members is that the new approach is well-motivated and supported by extensive theoretical and empirical analysis and that it addresses a meaningful gap in the existing literature. These points are key arguments for accepting the paper:
* S1: The authors provide a detailed time complexity analysis for the new method from different perspectives. The method approximates the temporal evolution by piecewise linear interpolations with theoretical analysis, which is a novel approach.
* S2: Experiments are comprehensive, and the new method outperforms the baselines significantly. Experiments include both real and synthetic datasets.

Additionally, the authors successfully addressed key points raised by PC members.

---

### Decision · Program_Chairs · 2022-11-23

Accept (Poster)